# Chemoselectivity change in catalytic hydrogenolysis enabling urea-reduction to formamide/amine over more reactive carbonyl compounds

Takanori Iwasaki [1,2] ✉, Kazuki Tsuge[1,2], Naoki Naito [1] & Kyoko Nozaki [1] ✉

The selective transformation of a less reactive carbonyl moiety in the presence of more reactive ones can realize straightforward and environmentally benign chemical processes. However, such a transformation is highly challenging because the reactivity of carbonyl compounds, one of the most important functionalities in organic chemistry, depends on the substituents on the carbon atom. Herein, we report an Ir catalyst for the selective hydrogenolysis of urea derivatives, which are the least reactive carbonyl compounds, affording formamides and amines. Although formamide, as well as ester, amide, and carbamate substituents, are considered to be more reactive than urea, the proposed Ir catalyst tolerated these carbonyl groups and reacted with urea in a highly chemoselective manner. The proposed chemo- and regioselective hydrogenolysis allows the development of a strategy for the chemical recycling of polyurea resins.

Synthetic organic chemistry creates complex molecules by repeatedly selecting and converting one of the numerous chemical bonds in a molecule[1]. Thus, the selective transformation of an intended functional group in an organic molecule carrying multiple functionalities is a fundamental and indispensable subject in organic synthesis. When the functional group to be reacted has a higher reactivity than any of the other functional groups present in the reactant, selective transformation can be achieved under appropriate reaction conditions and by using appropriate reagents. In other words, the selective transformation of the less reactive functional group against the generally accepted reactivity orders remains an inherent issue in state-of-the-art organic synthesis[2–4].

Carbonyl groups are an important class of functional groups in organic chemistry and accept various nucleophiles at the carbonyl carbon to interconvert into different carbonyl compounds via nucleophilic substitution or to afford alcohols via nucleophilic addition[5]. Because the reactivity of the carbonyl carbon is controlled by the two substituents on the carbon atom, the relative reactivity of carbonyl

compounds is strictly defined by the innate nature of the substituents (Fig. 1a)[6]. To change the reactivity order, some approaches using pre-[7] or in situ protection[8] of reactive carbonyl groups have been established (Fig. 1b). For example, the conversion of an aldehyde to an acetal (**I**) is often conducted to protect carbonyl groups from nucleophiles[7]. Alternatively, in situ protection through the tentative addition of a nucleophile to convert a carbonyl carbon to an sp$^3$-hybridized carbon has been well studied[8]. In their pioneering work on in situ protection, Luche demonstrated that the combined use of NaBH$_4$ and one equivalent of CeCl$_3$ enables the selective reduction of ketones in the presence of aldehydes, which originates from the selective conversion of aldehydes to Ce-stabilized *gem*-diols (**II**) to resist the NaBH$_4$ nucleophile[9]. Steric protection is another method for in situ protection. Yamamoto showed that capping a sterically less bulky aldehyde moiety with a sterically demanding aluminum Lewis acid (**III**) enables the selective addition of organolithium reagents to ketones[10].

While the selection of ketones over aldehydes is realized by the protection of aldehydes, the discrimination of amides over

[1]Department of Chemistry and Biotechnology, Graduate School of Engineering, The University of Tokyo, 7-3-1 Hongo, Bunkyo-ku, Tokyo 113-8656, Japan.
[2]These authors contributed equally: Takanori Iwasaki, Kazuki Tsuge. ✉e-mail: iwasaki@chembio.t.u-tokyo.ac.jp; nozaki@chembio.t.u-tokyo.ac.jp

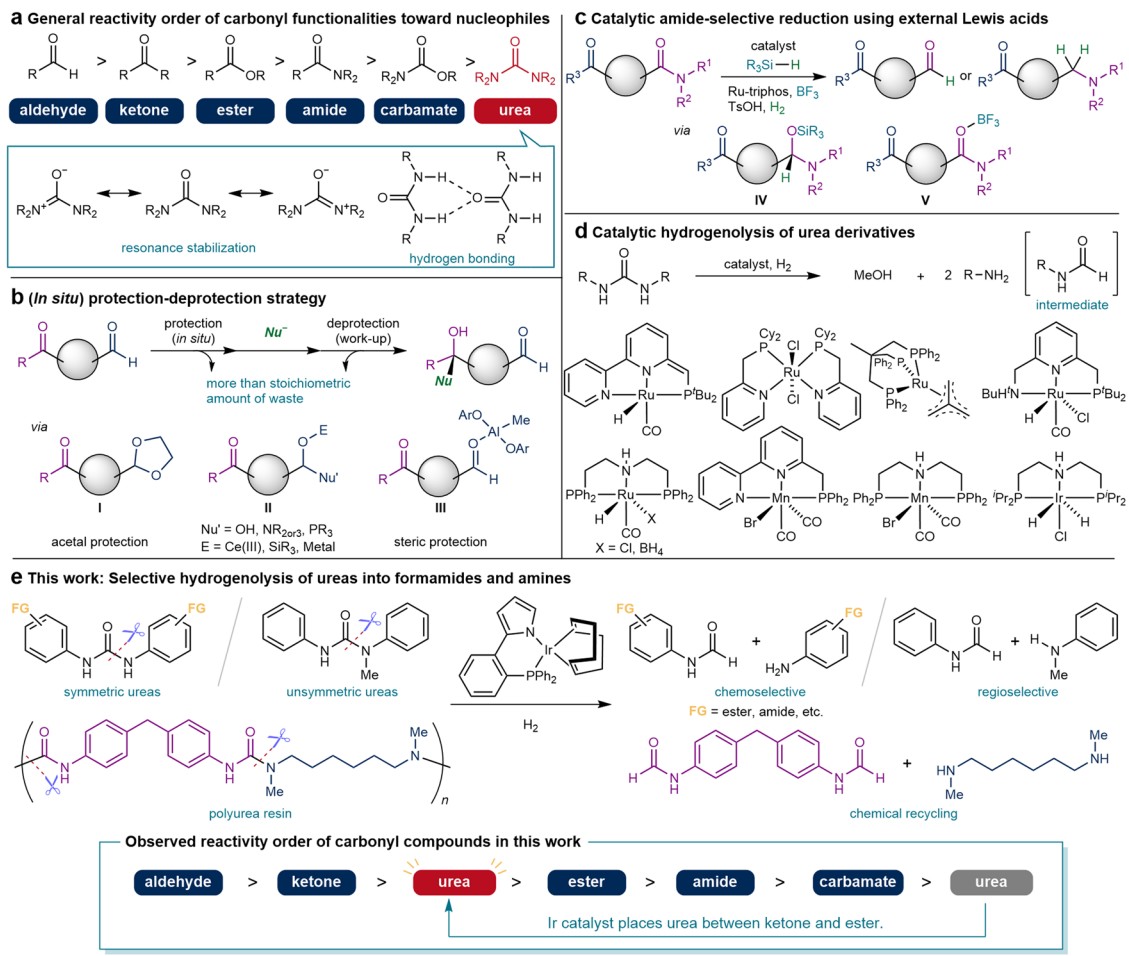

**Fig. 1 | Methods for the selective transformation of less reactive carbonyl compounds. a** Reactivity order of carbonyl compounds[27]. **b** Indirect synthetic approaches by protection-deprotection strategy. **c** Amide-selective catalytic reduction using hydrosilanes or dihydrogen. **d** Catalytic hydrogenolysis of urea functionality into methanol and two molecules of amines. **e** Chemoselective hydrogenolysis of urea derivatives into formamides and amines and unprecedented reactivity order of carbonyl compounds (This work).

ketones/esters relies on the high basicity of the amide oxygen[11–15]. Schwartz's reagent (Cp$_2$Zr(H)Cl) mediates the selective reduction of amides to aldehydes, even in the presence of more reactive but less basic esters[12]. The activation of amides by Tf$_2$O and subsequent reduction by Hantzsch esters afforded amines bearing ketone and ester functionalities[13]. When oxygen atoms can be trapped by silicon or boron atoms, selective catalytic reduction of amides over aldehydes/ketones/esters[16–20] or ketones over aldehydes[21] can be accomplished by taking advantage of the strong Si−O or B−O bonds (Fig. 1c). For example, Tinnis and Adolfsson demonstrated the reduction of amides to aldehydes using a Mo catalyst and tetramethyldisiloxane[16]. In this reaction, silylated aminal (**IV**) is proposed as an intermediate; thus, reduction can selectively afford the aldehyde as the product. The selective reduction of amides to amines has also been achieved using metal catalysts and hydrosilanes as reducing reagents[16–19]. In these reactions, silanes act as not only reducing reagents but also Lewis acids that selectively activate more basic amide bonds. Similarly, by adding BF$_3$ as a Lewis acid, Ru-triphos reportedly catalyzed the selective hydrogenation of amides over esters via a boron adduct (**V**)[20]. Although catalyst-controlled changes in chemoselectivity without external Lewis acidic additives provide more straightforward synthetic strategies[22], as exemplified by the chemoselectivity control of nitrogen and oxygen nucleophiles[2–4,23–26], it is still highly challenging, especially for carbonyl compounds.

Among carbonyl compounds, urea is the least reactive[27] because of stabilization by resonance from the lone pair on the nitrogen atoms to the carbonyl carbon and strong intermolecular hydrogen bonding (Fig. 1a). Therefore, the hydrogenolysis of urea in the presence of other carbonyl functionalities is one of the most challenging and unsolved chemoselectivities[28]. In fact, Milstein reported that a competitive reaction of urea and formamide using a Ru catalyst resulted in the selective hydrogenolysis of formamide[29]. Since Milstein reported the Ru-catalyzed hydrogenolysis of urea derivatives into two molecules of amine and methanol in 2011[29], several catalytic systems, including Ru[30–35], Mn[35–37], and Ir[38], have been used for the same conversion (Fig. 1d). As formamide intermediates are more reactive than urea, urea derivatives are fully reduced to amines and methanol under these catalytic systems, with the exception using a Ru-triphos catalyst to give a mixture of formamides and amines from different diaryl- and dialkylureas[31,39]. In this catalytic system, however, not only urea but also ester, amide, and other carbonyl functionalities were hydrogenated under similar conditions (*vide infra*)[31]. Indeed, urea-selective hydrogenolysis in the presence of more reactive carbonyl functionalities, such as esters, has never been achieved using these catalysts and is believed to be unfeasible[29–39].

Herein, we report the Ir-catalyzed chemoselective hydrogenolysis of urea derivatives into formamides and amines using hydrogen gas (Fig. 1e), where the urea functionality was selectively reduced even in the presence of formamide intermediates and more reactive carbonyl

functionalities such as esters and amides (FG = COOEt, CON$^n$Pr$_2$, etc.). When unsymmetric ureas were employed, the regioselective cleavage of the C–N bond was achieved. Mechanistic studies highlighted two possible reaction mechanisms for the origin of the unique chemoselectivity: (1) metal-ligand cooperative proton and hydride transfer to urea derivatives through the selective protonation of the more basic carbonyl oxygen by a proton from the ligand; and (2) thermal decomposition of urea derivatives into amines and isocyanates, the latter being selectively hydrogenated to formamide by the Ir catalyst. The selective hydrogenolysis of urea into formamide and amine serves as a strategy for the chemical recycling of polyurea resins by the transfer of molecular hydrogen.

## Results and discussion
### Catalyst screening
We initiated our study using 1,3-diphenylurea (**1a**) as the representative substrate and explored different catalysts under a hydrogen atmosphere (2 MPa) in toluene at 130 °C (Fig. 2). With Ir complex **4** bearing the phosphine-pyrrolido **5** ligand, the hydrogenolysis of **1a** occurred to afford formanilide (**2a**) and aniline (**3a**) in 82 and 83% yields, respectively. The selectivity of urea hydrogenolysis over further reduction of formanilide (**2a**) was calculated to be 99%, showing excellent chemoselectivity of catalyst **4**. When Ir complex **6** bearing benzimidazole, which has a more acidic N–H bond than pyrrole, as the coordinating site was employed, 77% of **1a** was reduced to give **2a** and **3a** both in 77% yields with excellent selectivity. In contrast to catalysts **4** and **6** bearing *N*-heterocycles, replacing the *N*-heterocyclic coordinating site with sulfonato (**7**) or carboxylato (**8**) groups resulted in low yields of both **2a** and **3a** because of the low conversion of **1a**. Analogous Rh complex **9** carrying the phosphine-pyrrolido **5** ligand allowed very low conversion. Almost no reaction occurred with representative Rh and Ir hydrogenation catalysts such as Wilkinson's catalyst **10** and Vaska's catalyst **11**. Crabtree's catalyst **12** afforded **2a** and **3a** in low yields with a low selectivity of 62%. No reactions occurred without a catalyst.

### Substrate scope of symmetric urea derivatives
After optimizing the reaction conditions with **4** as the best catalyst (Supplementary Tables 1–2), we found two optimal conditions, as shown in Fig. 3a. The reaction under reduced H$_2$ pressure (1 MPa) in THF at 130 °C (Condition A) afforded **2a** and **3a** in 80 and 82% yields, respectively, with excellent selectivity. Alternatively, the reaction using a catalytic amount of KO$^t$Bu as an additive in toluene (Condition B) showed higher catalytic efficiency to give **2a** and **3a** in 83 and 114%

yields, with a slightly decreased selectivity due to the over-reduction of **2a** to **3a**.

With the optimized conditions in hand, the substrate scope of the hydrogenolysis was investigated (Fig. 3a; see also Supplementary Figs. 4–6). When 1,3-diarylureas bearing electron-withdrawing halogen moieties, F (**1b**) and Cl (**1c**), at the *para*-position were employed in the reaction, the corresponding products were obtained in high yields and selectivities. Steric hindrance had a negligible effect; *meta-* and *ortho*-substituted 1,3-diphenylureas **1d** and **1e** underwent hydrogenolysis to give the corresponding products, which is in sharp contrast to previous results that are largely affected by steric hindrance[36]. Bromo-substituted 1,3-diphenylurea **1f** required 10 mol% catalyst loading for full conversion and resulted in over-reduction to give 42% yield of **2f** along with 135% yield of **3f**. Notably, no C–Br bond cleavage was observed under these conditions.

Electron-donating substituents such as alkyl, methoxy, and dimethylamino groups (**1g**–**1i**) slightly affected the reaction efficiency, consistent with the general electronic demand for nucleophilic addition to carbonyl compounds. Notably, the present catalyst selectively cleaved the C–N bond of the urea moiety, even for 1,3-diarylurea **1j** possessing an ester moiety. This unprecedented chemoselectivity was further confirmed by the competitive reaction of **1a** with ethyl benzoate (**13**), where selective hydrogenolysis of **1a** took place, and >99% of **13** was recovered without any loss of the ester group (Fig. 3b). These results clearly show that the present catalyst possesses unique chemoselectivity that has not been reported in the literature, and that the innate reactivity order of carbonyl moieties can be reversed upon the addition of a catalytic amount of an exogenous control element. Similarly, the competitive reaction of **1a** with carbamate **14** resulted in the selective hydrogenolysis of urea **1a**, indicating that carbamate is less reactive than urea in the present reaction (Fig. 3b). In contrast, no selectivity was observed for the competitive reaction between urea **1a** and a ketone, giving rise to a mixture of **2a**, **3a**, and alcohol (Supplementary Fig. 5). Therefore, the present catalyst places ureas between ketones and esters in its reactivity order, in contrast to the conventional reactivity order (Fig. 1a). The absence of observed reactions of **13** and **14** along with the over-reduction of formamides **2** in some cases suggests that the present catalyst **4** reduces formamides **2** more easily than ester **13** and carbamate **14**. The amide and cyano moieties, which are believed to have higher reactivity than the urea group, were also tolerated under the reaction conditions (**1k** and **1l**). The urea derivatives of primary and secondary alkylamines (**1m** and **1n**) underwent

**Fig. 2 | Catalyst screening for the chemoselective hydrogenolysis of 1,3-diphenylurea (1a).** Reaction conditions: **1a** (0.17 mmol) and catalyst (3 mol%) in toluene (2 mL) under H$_2$ (2 MPa) at 130 °C for 18 h. Yields were determined by $^1$H NMR analysis relative to an internal standard. Yield of amine **3a** was reported based on the mole of **1a**, being 200% at maximum. n.d.: not detected. Selectivity value = yield of **2a**/{(yield of **2a** + yield of **3a**)/2} × 100 (%).

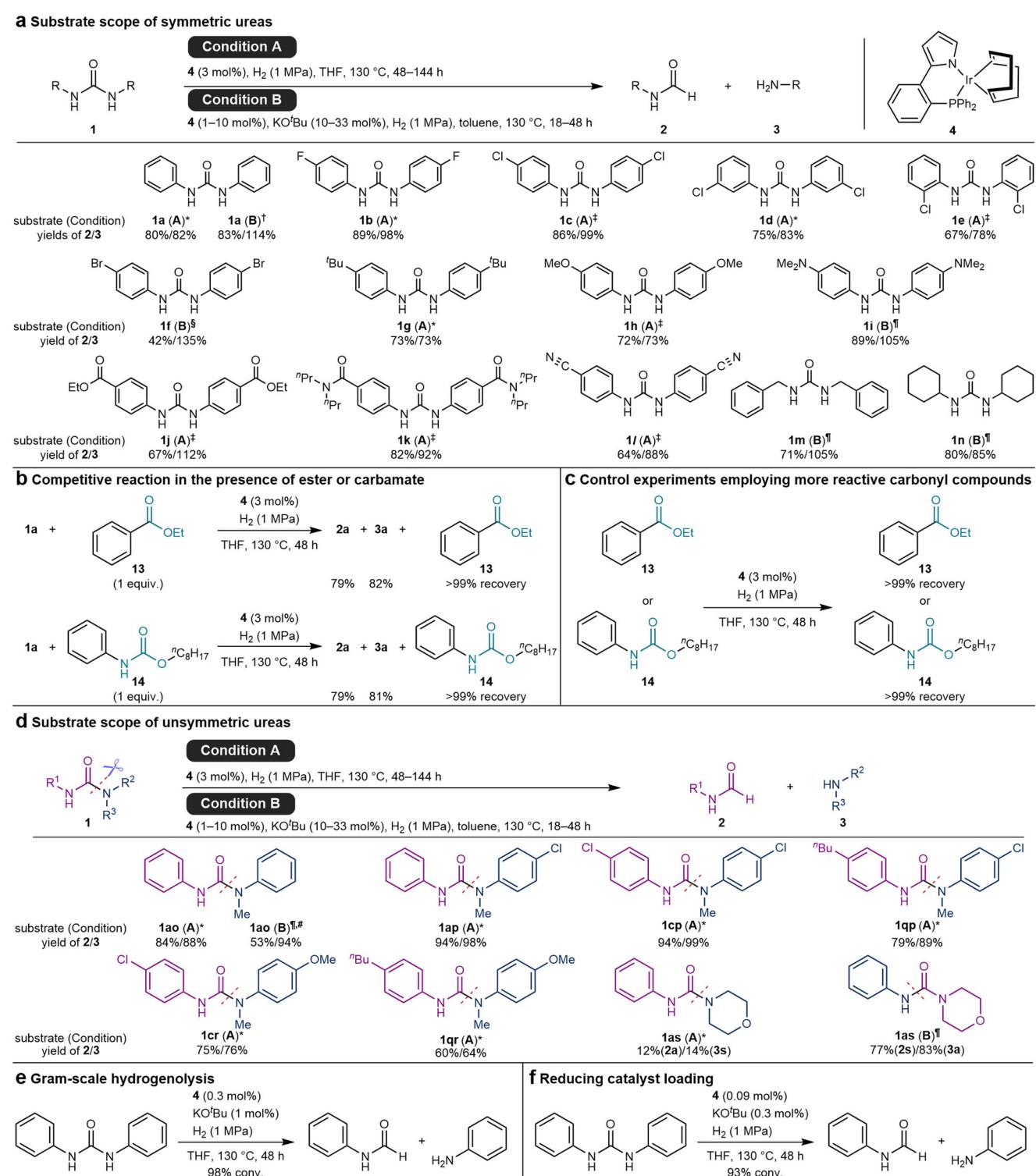

**Fig. 3 | Scope of chemoselective hydrogenolysis of urea derivatives.** All yield values were determined by ¹H NMR using an internal standard. Yield of amine **3** was reported based on the mole of urea **1**, being 200% at maximum. **a** Substrate scope of symmetric urea derivatives. *, Reaction for 48 h; †, **4** (1 mol%) and KO$^t$Bu (10 mol%) for 18 h; ‡, Reaction for 144 h; §, **4** (10 mol%) and KO$^t$Bu (33 mol%) for 48 h; ¶, **4** (3 mol%) and KO$^t$Bu (10 mol%) for 48 h.

**b** Competitive reaction of **1a** with ester **13** or carbamate **14**. **c** Control experiments employing more reactive carbonyl compounds, ester **13** and carbamate **14**. **d** Substrate scope of regioselective hydrogenolysis of unsymmetric urea derivatives. *, Reaction for 48 h; ¶, **4** (3 mol%) and KO$^t$Bu (10 mol%) for 48 h; #, Along with 39% yield of **3a**. **e** Gram-scale hydrogenolysis. **f** Reducing catalyst loading.

**a Metal-ligand cooperative hydrogenolysis of urea**

**b Thermal decomposition of urea into isocyanate and its hydrogenation**

**Fig. 4 | Possible reaction pathways. a** Metal-ligand cooperative pathway. **b** Thermal decomposition pathway. For experimental supports and detailed discussions, see the Supplementary Information.

hydrogenolysis with the aid of the KO$^t$Bu additive (Condition B) and afforded the products in high yields and moderate selectivities.

### Functional group tolerance

Control experiments using ester **13** or carbamate **14** as substrates under the optimized conditions revealed that the present catalytic system did not convert these functionalities (Fig. 3c). These results clearly differentiate our catalytic system from that using the Ru-triphos catalyst, which exhibits catalytic activity toward the hydrogenolysis of not only urea but also other carbonyl compounds such as esters, amides, and carbonates[31].

### Regioselective hydrogenolysis of unsymmetric urea derivatives

Next, we focused on the hydrogenolysis of unsymmetric urea derivatives (Fig. 3d). Because two C–N bonds are cleaved in the hydrogenolysis of urea derivatives in previous reports[29–38], the regioselectivity of the first C–N bond cleavage in unsymmetric ureas has not yet been addressed. When a methyl group is introduced onto the one nitrogen atom of 1,3-diphenylurea (**1ao**), the hydrogenolysis regioselectively occurred at the C–N bond of the secondary amine to give formanilide (**2a**) and *N*-methylaniline (**3o**) in 84 and 88% yields, respectively, along with small amounts of **3a** (3%) under Condition A. Although exchange reaction between an amide and an amine occasionally takes place under harsh conditions[7], the reaction between products **2a** and **3o** was negligible for urea **1ao** under Condition A (Supplementary Fig. 19). When **1ao** was subjected to Condition B, **2a** and **3o** were produced in 53 and 94% yields, respectively, along with 39% of **3a** probably via the over-reduction of **2a**. The introduction of an electron-withdrawing group (Cl) into the *N*-methylaniline moiety (**1ap**) or both aryl groups (**1cp**) did not affect the yield and regioselectivity, affording formanilides (**2a** or **2c**) and *N*-methyl-4-chloroaniline (**3p**) in a regioselective manner. Electron-donating groups such as $^n$Bu and OMe at the *para*-position of either the aniline or *N*-methylaniline moiety (**1qp** and **1cr**) decreased the reaction efficiency but did not affect the regio- and chemoselectivities. Lower yields were obtained following the introducing of electron-donating groups at the *para*-position of both aryl groups (**1qr**).

The reaction of unsymmetric urea **1as** consisting of aniline and morpholine moieties resulted in the formation of **2a** and morpholine (**3s**) in 12% and 14% yields, respectively, with 17% conversion of **1as** under Condition A. In contrast, under Condition B, the conversion of

**1as** was significantly improved to >99%, and the regioselectivity was completely inverted to afford **2s** and **3a** in 77 and 83% yields, respectively (see the Supplementary Information, Section 1-11).

### Hydrogenolysis of urea in gram-scale and reducing catalyst loading

The gram-scale hydrogenolysis of **1a** with 0.3 mol% **4** and 1 mol% KO$^t$Bu in THF afforded **2a** and **3a** in 93 and 105% yields, respectively (Fig. 3e). Further reducing the catalyst loading to 0.09 mol% resulted in the almost full conversion of **1a**, which corresponds to a turn-over number of 1033 (Fig. 3f), a value more than 5-times higher than that previously reported for the hydrogenolysis of urea into amine and methanol[32].

### Reaction pathways

We postulated two possible reaction pathways, as shown in Fig. 4, based on mechanistic studies (for experiments and detailed discussions, see the Supplementary Information). One pathway involves metal-ligand cooperation[40,41] in the hydrogenolysis process (Fig. 4a). Upon treating Ir precatalyst **4** with hydrogen gas, Ir intermediate **A** is formed. The heterolytic cleavage of H$_2$ by the Ir–N bond forms complex **B**. The protonation of the carbonyl oxygen by the acidic N–H bond in the pyrrole moiety of **B** discriminates the more basic carbonyl oxygen in ureas than that in formamides (**C**). This is the origin of the unprecedented chemoselectivity of the present catalytic system. Subsequent hydride transfer from the Ir center or concomitant transfer of the proton and hydride to the carbonyl C=O bond forms intermediate **D** and regenerates Ir complex **A**. The elimination of one amino group from intermediate **D** yields an amine and a formamide in a selective manner.

In this catalytic cycle, the Ir center and pyrrole moiety cooperatively reduced the C=O bond of the urea moiety. Although such metal-ligand cooperation[40,41] has been well established for the catalytic reduction of carbonyl compounds, including urea derivatives, the proposed catalytic cycle involves a unique mechanism, namely, the nitrogen atom in the pyrrole ring, which directly participates in the heterolytic cleavage of H$_2$ as well as the proton transfer step.

Another possibility is the thermal decomposition of urea into isocyanate **F** and amine via zwitterion **E** prior to hydrogenation by the Ir catalyst. Isocyanate **F**, which is more electrophilic than formamide, is reduced by the Ir catalyst to form the formamides[39,42,43] (Fig. 4b).

**Fig. 5 | Catalytic hydrogenolysis of polyurea resin 17.** All yield values were determined by [1]H NMR using an internal standard. PDI: poly dispersity index. See the Supplementary Information for details.

Control experiments and kinetic studies were consistent with these two reaction pathways, and we could not rule out one of the two possible pathways at this moment (see the Supplementary Information, Sections 1-5–1-11).

## Chemical recycling of polyurea resins

Finally, chemoselective hydrogenolysis was applied to the degradation of polyurea resins for chemical recycling[44,45] (Fig. 5). Degradations of polyurea resins for chemical recycling by hydrogenolysis[35,38,46,47], hydrolysis[48], and solvolysis[49,50] have been reported, but there are still room to improve (see the Supplementary Information, Section 1-11). Contrary, the present chemo- and regioselective catalytic hydrogenolysis enables easy separation of degraded products and realizes a novel chemical recycling strategy. The condensation reaction of diisocyanate **15** with diamine **16** afforded polyurea resin **17** with $M_n = 64 \times 10^3$ as a less soluble off-white solid. The hydrogenolysis of **17** with a catalytic amount of **4** and KO[t]Bu under slightly modified conditions mainly afforded diformamide **18** (72% yield) and diamine **16** (88% yield), the latter being one of the monomers in the formation of polyurea. The diphenylmethane unit was recovered as diformamide **18** in 72% yield, along with monoformamide **19** (24%) and diamine **20** (2%). This result is consistent with the regioselectivity observed in the hydrogenolysis of unsymmetric ureas, as exemplified by that of **1ao**, and the formation of **19** and **20** could be explained by over-reduction. Notably, the obtained diformamide **18** still possesses two carbonyl groups at both ends, in contrast to the previously reported degradation of polyurea resins via hydrogenolysis[35,38,46,47] that lost the carbonyl groups. Because polyurea resins can be synthesized by the dehydrogenative coupling of formamides with amines[51], the combination of the present chemo- and regioselective hydrogenolysis and dehydrogenative coupling enables the chemical recycling of polyurea resins consisting of two alternating different diamine segments, as in polyurea **17**, only by the transfer of molecular hydrogen (see the Supplementary Information, Section 1-12). In addition, because urea, ester, and carbamate are often found in polymer materials as polyureas, polyesters, and polyurethanes, respectively, the exceptionally high chemoselectivity of the proposed catalytic system is promising for the selective chemical recycling of polyureas from mixed polymer materials.

In conclusion, the Ir catalytic system allows the selective hydrogenolysis of one C–N bond in the urea functionality to afford formamides and amines as products. In addition to formamides, reactive carbonyl functionalities, such as esters, amides, and carbamates, are well tolerated under the reaction conditions. We also demonstrated the hydrogenative degradation of polyurea resins using the proposed catalytic system, in which the carbonyl carbon was retained in the degraded monomer.

The change in chemoselectivity demonstrated herein provides a new strategy for atom-economical and environmentally benign processes, unlike existing strategies that rely on stoichiometric reagents, for the selective transformation of carbonyl compounds. The combination of the proposed catalysis and dehydrogenative coupling reactions allows chemical recycling via the transfer of molecular hydrogen.

## Methods
### General procedure for the hydrogenolysis of urea derivatives using Ir catalyst 4 (Condition A)

A 50 mL stainless steel autoclave, a glass tube, and a stirring bar were dried in an oven at 150 °C, and then cooled inside a glovebox under argon atmosphere. Catalyst **4** (9.4 mg, 15 μmol), urea (0.50 mmol), and THF (3 mL) were added into the glass tube, and the tube was capped with a funnel to prevent evaporation of the solvent. After the glass tube was set in the autoclave, the autoclave was sealed. The autoclave was brought out from the glovebox. The autoclave was degassed three times using $H_2$ and was pressurized with 1 MPa of $H_2$ for 5 min with stirring. The reaction mixture was stirred in an isothermal heating block at 130 °C for 48 h. After cooling to room temperature, $H_2$ was vented off carefully. To the reaction mixture, DMSO-$d_6$ (ca. 2 mL) was added to homogenize the reaction mixture, and heptane (25.1 mg, 0.25 mmol) or dibromomethane (86.9 mg, 0.50 mmol) was added as an internal standard. The solution (THF/DMSO-$d_6$ = 3/2, 0.3 mL) was transferred into an NMR tube and diluted with DMSO-$d_6$ (0.3 mL) for NMR analysis. The conversion of urea and the yields of formamide and amine were determined by [1]H NMR spectroscopy with the internal standard (**1d**: heptane, others: dibromomethane). To the combined reaction mixture, ethyl acetate (30 mL) was added, and the solution was extracted with 1 M HCl aq. (20 mL × 3). The combined aqueous layers were neutralized with $NaHCO_3$ and extracted with $CH_2Cl_2$ (50 mL × 3). The combined $CH_2Cl_2$ layers were washed with brine (100 mL), dried over anhydrous $Na_2SO_4$, and filtered. After adding 1 M HCl in MeOH, the solution was stirred at room temperature for 3 h and concentrated under reduced pressure to obtain the corresponding anilinium chloride salt. The ethyl acetate layer was washed with brine, dried over anhydrous $Na_2SO_4$, filtered, and concentrated under reduced pressure. The crude residue was purified by silica gel column chromatography with $CH_2Cl_2$/ethyl acetate as an eluent to obtain formamide.

### General procedure for the hydrogenolysis of urea derivatives using Ir catalyst 4 in the presence of KO[t]Bu (Condition B)

A 50 mL stainless steel autoclave, a glass tube, and a stirring bar were dried in an oven at 150 °C, and then cooled inside a glovebox under argon atmosphere. Catalyst **4** (9.4 mg, 15 μmol), urea (0.50 mmol), KO[t]Bu (5.6 mg, 0.05 mmol), and toluene (3 mL) were added into the glass tube, and the tube was capped with a funnel to prevent evaporation of the solvent. After the glass tube was set in the autoclave, the autoclave was sealed. The autoclave was brought out from the glovebox. The autoclave was degassed three times using $H_2$ and was pressurized with 1 MPa of $H_2$ for 5 min with stirring. The reaction mixture was stirred in an isothermal heating block at 130 °C for 48 h.

After cooling to room temperature, H₂ was vented off carefully. To the reaction mixture, DMSO-$d_6$ (ca. 2 mL) and 0.5 M HCl aq. (0.1 mL) were added to homogenize and neutralize the reaction mixture, and dibromomethane (86.9 mg, 0.50 mmol) was added as an internal standard. The solution (toluene/DMSO-$d_6$ = 3/2, 0.3 mL) was transferred into an NMR tube and diluted with DMSO-$d_6$ (0.3 mL) for NMR analysis. The conversion of urea and the yields of formamide and amine were determined by $^1$H NMR spectroscopy with the internal standard. To the combined reaction mixture, ethyl acetate (30 mL) was added, and the solution was extracted with 1 M HCl aq. (20 mL × 3). The combined aqueous layers were neutralized with NaHCO₃ and extracted with CH₂Cl₂ (50 mL × 3). The combined CH₂Cl₂ layers were washed with brine (100 mL), dried over anhydrous Na₂SO₄, and filtered. After adding 1 M HCl in MeOH, the solution was stirred at room temperature for 3 h and concentrated under reduced pressure to obtain the corresponding anilinium chloride salt. The ethyl acetate layer was washed with brine, dried over anhydrous Na₂SO₄, filtered, and concentrated under reduced pressure. The crude residue was purified by silica gel column chromatography with CH₂Cl₂/ethyl acetate as an eluent to obtain formamide.

## Hydrogenolysis of polyurea 17

A 50 mL stainless steel autoclave, a glass tube, and a stirring bar were dried in an oven at 150 °C, and then cooled inside a glovebox under argon atmosphere. Catalyst **4** (3.1 mg, 5 μmol), polyurea resin **17** (32.9 mg, 0.16 mmol/urea moiety), KO$^t$Bu (1.9 mg, 0.017 mmol), and THF (3 mL) were added into the glass tube, and the tube was capped with a funnel to prevent evaporation of the solvent. After the glass tube was set in the autoclave, the autoclave was sealed. The autoclave was brought out from the glovebox. The autoclave was degassed three times using H₂ and was pressurized with 1 MPa of H₂ for 5 min with stirring. The reaction mixture was stirred in an isothermal heating block at 130 °C for 18 h. After cooling to room temperature, H₂ was vented off carefully. To the reaction mixture, 0.5 M HCl aq. (33 μL) was added to neutralize the reaction mixture, and dibromomethane (57.9 mg, 0.33 mmol) was added as an internal standard. The solution (0.1 mL) was transferred into an NMR tube and added with DMSO-$d_6$ for NMR analysis. The yields of products were determined by $^1$H NMR spectroscopy with the internal standard (NMR yields were calculated by subtracting impurities including **17**). The combined reaction mixture was evaporated and extracted with hexane (3 mL × 6). The combined hexane layers were evaporated to obtain N,N'-dimethyl-1,6-diaminohexane (**16**) as a colorless liquid (5.8 mg, 48%). The residue was purified by silica gel column chromatography with CH₂Cl₂/ethyl acetate as an eluent to obtain N,N'-(methylenebis(4,1-phenylene))diformamide (**18**) as a white solid (13.8 mg, 65%).

## Data availability

The data generated in this study are available in the main text, Supplementary Information, or Supplementary Data. Source data are available for Supplementary Figs. 16, 25, 27–31, 33–37, 39–43, and 45–49 in the associated source data file via https://doi.org/10.6084/m9.figshare.22725836. X-ray crystal data can also be obtained from the joint Cambridge Crystallographic Data Centre (CCDC: 2225764 (**4**), 2225765 (**7**), 2225766 (**9**), and 2225767 (**23**)) via https://www.ccdc.cam.ac.uk/structures/. Source data are provided with this paper.

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

## Acknowledgements

A part of this work was conducted at Advanced Characterization Nanotechnology Platform of the University of Tokyo (JPMXP1222UT0196, –23UT0046), supported by Nanotechnology-Platform-Project by the Ministry of Education, Culture, Sports, Science & Technology (MEXT), Japan. The computations were carried out using the Research Center for Computational Science, Okazaki, Japan (Project: 22-IMS-C017, 23-IMS-C014). This work was partly supported by JST ERATO (No. JPMJER2103) to K.N., The Grant-in-Aid for Transformative Research Areas (A) JP21A204 in Digitalization-driven Transformative Organic Synthesis (Digi-TOS) from MEXT, Japan (No. JP22H05340) to T.I., The New Energy and Industrial Technology Development Organization (NEDO) (No. JPNP20004) to T.I., ENEOS TONENGENERAL foundation to T.I., and Sumitomo foundation to T.I.

## Author contributions

T.I., K.T., and N.N. performed experiments and analyzed data. T.I. and K.T. co-wrote the original draft. T.I., K.T., N.N., and K.N. reviewed and edited it.

## Competing interests

A Japan patent application on this work has been filed (Japan Patent Application No. 2022-020544), where K.N., T.I., K.T., and N.N. are listed as inventors.
