## [Peer Review File · Nature Communications]

Chemoselectivity Change in Catalytic Hydrogenolysis enabling Urea-reduction to Formamide/Amine over More Reactive Carbonyl CompoundsReviewers' Comments:

Reviewer #1:

Remarks to the Author:

In this manuscript, Iwasaki, Nozaki and co-workers have demonstrated a new pathway for the transformation of urea derivatives to formamides and amines via catalytic hydrogenolysis. Overall, the work is very interesting and have been carried out with appropriate standard by providing sufficient details. In my opinion the manuscript is suitable for publication in Nat Commun provided the following comments are addressed:

1. The background of polyurea recycling should be provided in the main manuscript with appropriate citations. I note that it has been provided in the ESI but should form a part of the introduction.
2. The authors should perform hydrogenolysis of N-substituted urea derivatives (on both nitrogen) e.g. 1,3-dimethyl-1,3-diphenylurea. This can provide evidence to support one of the pathways described in Figure 4 of the main manuscript as this urea derivative would not be able to form intermediate analogous to E in Figure 4. If the product is obtained, this could support pathway via metal-ligand cooperation.
3. I would also encourage to report the hydrogenolysis of urea derivatives containing aliphatic amines e.g. 1,3-dihexylurea, etc. According to the hypothesis provided for the chemoselectivity (page 7: The protonation of the carbonyl oxygen by the acidic N-H bond in the pyrrole moiety of B discriminates the more basic carbonyl oxygen in ureas than that in formamides. This is the origin of the unprecedented chemoselectivity of the present catalytic system), aliphatic urea derivatives should be more reactive as their carbonyl oxygen atom should be more basic than those of aromatic urea derivatives.

Reviewer #2:

Remarks to the Author:

In this manuscript, the chemoselective hydrogenation of ureas to formamides and amines using Ir-catalyst with PN-ligands is reported. The authors report, that this catalysts system is able to preferably hydrogenate urea functions, even in the presence of ester, amide and carbamate groups which is so far unprecedented as usually the urea function is usually more difficult to hydrogenate than the later given. This is a remarkable achievement in terms of selective hydrogenations of carbonyl compounds. The system was applied to a series of symmetric as well as unsymmetric ureas with several functionalities to be tolerated (Ester, amide, nitrile, aryl-halides). On top, also the hydrogenation of one polyurea was investigated. The work seems in principle very well executed and the (large) supporting information describes all experiments and products in a proper and detailed way. Also, a lot of experiments in trying to evaluate the mechanism (incl. kinetics) are provided. Therefore, the system and work presented is in principle of high interest and would fit to the broad readership of Nature Catalysis. But nevertheless, taking the high standards of the journal into account, certain points should be addressed in a revision, to further improve the manuscript to be recommend for acceptance.

Line 67-70. Yes, in ref 31 a mixture of the formamide and the aniline from diphenylurea is obtained with the Ru-triphos system. But more or less the same mixture is obtained in this work, in the hydrogenation of diphenyl urea (example 1a) and conditions/yields are not that different. So not really a big difference in that case and therefore this observation in selectivity of a ureas hydrogenation is not new. But for sure the system in ref 31 will not be compatible with esters and amides, as in ref 31 the Ru-triphos is also used to hydrogenate these carbonyl compounds. May the authors should specify it here a bit more (see lines 70-72) pointing out this drawback of the system in 31.

Figure 1E, bottom: Where in this scale would the author put the formamides? They were also hydrogenated with this catalyst to a certain extent (see yields of amine higher 100% as well as

polyurea hydrogenation). As shown in the SI, ureas is nevertheless hydrogenated preferably (see hydrogenations of urea in the presence of a formamide). May put them between urea and ester.

Line 115: Any explanation for the overreduction in relation to the two proposed mechanisms? Would overreduction be favored in the isocyanate path or the other? Or may any correlation to the water content and it's hydrolysis of the isocyanate and not overreduction? Was any exp. mad with just adding a bit of water to see, if the amount of aniline increases compared to formamide? By the way, please clearly indicate in figure 3A, that the yields above 100% are caused by overreduction. Without any indication there or comment in the caption, the yields higher 100% are looking a bit odd for the reader on the first view.

Line 186-187, lines 200-201 as well as figures 3B, S6 and S15: Generally, I would recommend a in dept computational study on the mechanism, which could help to evaluate, which is the most likely pathway. But this would require a lot of computational work and would be too much for a revision, but I recommend addressing this in a follow-up work. Nevertheless, I strongly recommend for this manuscript to also make the NBO analysis for the methyl-substituted urea and cyclic urea in S6 as well as the ester and amide in figure 3B. If in all the other cases the electron density on the O is lower than in 1a, this would be a strong support for the metal-ligand cooperativity pathway. On the other hand, as the methylated urea 1o in scheme S6 is not really working, this is also an indication for the isocyanate path, as this substrate can't from the isocyanate at all (but 1t could do so, but is also not working).

Line 207-226 and figure 5: The authors used here conditions B from figure 3 which did result in significant overreduction (24% of 19) and therefore selectivity loss. As given in table S5, also reduction without added bases was tried (entry 1 and 2 in table S5), ut did not result in any product. Did the authors have any idea why, as in figure 3D, reduction worked also well with most methylated ureas under conditions A without the base. Also, it would be good to test at least some more polyureas, especially the more common ones with the non-methylated urea units (see for example the polyureas as given in ref 38 but also see: Polyureas, Ullmann's Encyclopedia of Industrial Chemistry, 2016, DOI: 10.1002/14356007.d21_d01.pub2). In terms of a proper chemcycling, the given example is not the most suitable one du to the 3:1 mixture of bisformamide : formamide obtained. Investigation of the hydrogenation performance on 2-3 other non-methylated polyureas would be very interesting for the reader in terms of polymer recycling.

REVIEWER COMMENTS

Reviewer #1 (Remarks to the Author):

In this manuscript, Iwasaki, Nozaki and co-workers have demonstrated a new pathway for the transformation of urea derivatives to formamides and amines via catalytic hydrogenolysis. Overall, the work is very interesting and have been carried out with appropriate standard by providing sufficient details. In my opinion the manuscript is suitable for publication in Nat Commun provided the following comments are addressed:

Comment 1: The background of polyurea recycling should be provided in the main manuscript with appropriate citations. I note that it has been provided in the ESI but should form a part of the introduction.

Response: According to the valuable suggestion by the reviewer, we included a brief background of polyurea degradation for chemical recycling together with the appropriate citations in the subsection titled “Chemical recycling of polyurea resins” in the revised manuscript. Because the contexture of the introduction mainly focuses on changing chemoselectivity in the transformations of carbonyl compounds, the description was not included in the introductory part. We also provided more detailed discussions on reported chemical degradation of polyurea resins in the Supplementary Information (Sections 1-12).

Comment 2: The authors should perform hydrogenolysis of N-substituted urea derivatives (on bot nitrogen) e.g. 1,3-dimethyl-1,3-diphenylurea. This can provide evidence to support one of the pathways described in Figure 4 of the main manuscript as this urea derivative would not be able to form intermediate analogous to E in Figure 4. If the product is obtained, this could support pathway via metal-ligand cooperation.

Response: We have performed hydrogenolysis of 1,3-dimethyl-1,3-diphenylurea (**1o**) and presented the results in the Supplementary Information (Fig. S6A, also see below Figure I). Fig. S6 is mentioned in the main manuscript in the subsection titled “Substrate scope of symmetric urea derivatives”.

Hydrogenolysis of **1o** did not occur under Condition A, and >99% of **1o** was recovered. Under Condition B, 16% of **1o** was consumed, and 22% of *N*-methylaniline (**3o**) was obtained. As pointed out by the reviewer, hydrogenolysis of **1o** would be a strong reason to rule out the “Thermal decomposition pathway”, which involves proton transfer from one nitrogen to the other. However, the results shown in Fig. S6A in our opinion are not convincing enough to reject this reaction pathway.

A Hydrogenolysis of tetra-substituted urea
Figure I. Hydrogenolysis of tetrasubstituted urea **1o** (from Fig. S6A).

Comment 3: I would also encourage to report the hydrogenolysis of urea derivatives containing aliphatic amines e.g. 1,3-dihexylurea, etc. According to the hypothesis provided for the chemoselectivity (page 7: The protonation of the carbonyl oxygen by the acidic N–H bond in the pyrrole moiety of **B** discriminates the more basic carbonyl oxygen in ureas than that in formamides. This is the origin of the unprecedented chemoselectivity of the present catalytic system), aliphatic urea derivatives should be more reactive as their carbonyl oxygen atom should be more basic than those of aromatic urea derivatives.

Response: We subjected two ureas containing aliphatic amine groups, namely 1,3-dibenzylurea (**1m**) and 1,3-dicyclohexylurea (**1n**), to hydrogenolysis as shown in Fig 3A (see also Figure II). Good yields were observed under Condition B, but the reactions were sluggish compared to 1,3-diarylureas under Condition A.

Figure II. Hydrogenolysis of ureas containing aliphatic amine groups (from Fig. 3A)

As pointed out by the reviewer, aliphatic amino groups improve the basicity of the urea oxygen to facilitate protonation by pyrrole in **B** (Fig. 4A). However, the electrophilicity of the carbonyl carbon is concurrently reduced by the electron-donating alkyl groups, thus affecting hydride transfer from Ir to the carbonyl carbon. Therefore, the observed lower reactivity of **1m** and **1n** is not inconsistent with the proposed reaction mechanism.

Reviewer #2 (Remarks to the Author):

In this manuscript, the chemoselective hydrogenation of ureas to formamides and amines using Ir-catalyst with PN-ligands is reported. The authors report, that this catalysts system is able to preferably hydrogenate urea functions, even in the presence of ester, amid and carbamate groups which is so far unprecedented as usually the urea function is usually more difficult to hydrogenate than the later given. This is a remarkable achievement in terms of selective hydrogenations of carbonyl compounds. The system was applied to a series of symmetric as well as unsymmetric ureas with several functionalities to be tolerated (Ester, amide, nitrile, aryl-halides). On

top, also the hydrogenation of one polyurea was investigated. The work seem in principle very well executed and the (large) supporting information describes all experiments and products in a proper and detailed way. Also, a lot of experiments in trying to evaluate the mechanism (incl. kinetics) are provided. Therefore, the system and work presented is in principle of high interest and would fit to the broad readership of Nature Catalysis. But nevertheless, taking the high standards of the journal into account, certain points should be addressed in a revision, to further improve the manuscript to be recommend for acceptance.

Comment 1: Line 67-70. Yes, in ref 31 a mixture of the formanilide and the aniline from diphenylurea is obtained with the Ru-triphos system. But more or less the same mixture is obtained in this work, in the hydrogenation of diphenyl urea (example 1a) and conditions/yields are not that different. So not really a big difference in that case and therefore this observation in selectivity of a ureas hydrogenation is not new. But for sure the system in ref 31 will not be compatible with esters and amides, as in ref 31 the Ru-triphos is also used to hydrogenate this carbonyl compounds. May the authors should specify it here a bit more (see lines 70-72) pointing out this drawback of the system in 31.

Response: We agree with this comment and modified the sentence as below.

“As formamide intermediates are more reactive than urea, urea derivatives are fully reduced to amines and methanol under these catalytic systems, with one exceptional case using a Ru-triphos catalyst to give a mixture of formanilide and aniline from 1,3-diphenylurea³¹. In this catalytic system, however, not only urea but also ester, amide, and other carbonyl functionalities were hydrogenated under similar conditions (vide infra)³⁹. Indeed, urea-selective hydrogenolysis in the presence of more reactive carbonyl functionalities, such as esters, has never been achieved using these catalysts and is believed to be unfeasible²⁹⁻³⁹.”

We also added reference 39 to the revised manuscript. This paper by Li *et al.* was published after our initial submission, and it expanded the substrate scope of the Ru-triphos catalytic system originally reported by Klankermayer, Leitner, and coworkers (reference 31), yet, the compatibility of ester, amide, and other carbonyl functionalities was not documented at all in reference 39.

In the revised manuscript, we remarked on this point in the main text and Note Section in order to clarify the novelty of our catalytic system as follows:

“After submitting this paper, another study was published that applied the Ru-triphos catalyst to a series of ureas, but again, chemoselectivity against other carbonyl compounds was not documented there.”

Comment 2: Figure 1E, bottom: Where in this scale would the author put the formamides? They were also hydrogenated with this catalyst to a certain extent (see yields of amine higher 100% as well as polyurea hydrogenation). As shown in the SI, ureas is nevertheless hydrogenated preferably (see hydrogenations of urea in the presence of a formamide). May put them between urea and ester.

Response: We agree with the reviewer that these particular formamides showed higher reactivity than esters in

the present catalytic system. Particularly, over-reduction of formamides was observed in some substrates in Figs. 3A and 3D-F. In addition, hydrogenation of *p*-fluoroformanilide (**2b**) afforded *p*-fluoroaniline (**3b**) in 8% yield (Fig, S11). Although formamides furnished much lower reactivity than ureas in this catalytic system, the observed reduction of formamides is in sharp contrast to the lack of reaction for ester **13** and carbamate **14** (Figs. 3B, C). According to reviewer's keen insight, we added this finding in the revised manuscript as a reference and note 40 as follows:

“The absence of observed reactions of **13** and **14** along with the over-reduction of formamides **2** in some cases suggests that the present catalyst **4** reduces formamides **2** more easily than ester **13** and carbamate **14**.”

Regarding the reactivity order shown in Fig. 1E, we did not change it because this order represents a general understanding.

Comment 3: Line 115: Any explanation for the overreduction in relation to the two proposed mechanisms? Would overreduction be favored in the isocyanate path or the other? Or may any correlation to the water content and it's hydrolysis of the isocyanate and not overreduction? Was any exp. mad with just adding a bit of water to see, if the amount of aniline increases compared to formamide?

Response: According to the comment, we conducted the reaction with 1 equiv of H₂O as an additive. The addition of H₂O decreased the selectivity from 99% to 61%, affording **2a** and **3a** in 42% and 95% yields, respectively (Figure III, top). Although the addition of H₂O to isocyanate forms **3a** through carbamic acid by decarboxylation, the hydrolysis of urea **1a** with H₂O can also produce **3a** via the same intermediate (Figure III, bottom). The suggested experiment clearly showed that dehydrated conditions are necessary for selective hydrogenolysis. This additional result and the following discussion were added to Table S2 as entry 4 in the revised Supplementary Information:

“When 1 equiv of H₂O was added to the conditions shown in entry 3, 42% of **2a** and a significant amount of **3a** (95%) were obtained, with a slightly lower conversion of 71% (entry 4). The formation of the significant amount of **3a** by adding H₂O is probably due to the hydrolysis of **1a** and/or phenyl isocyanate (**27a**) and subsequent decarboxylation.”

Figure III. The hydrogenolysis of **1a** in the presence of H₂O (Top) and possible explanation of the formation of **3a** by H₂O (bottom).

Comment 4: By the way, please clearly indicate in figure 3A, that the yields above 100% are caused by overreduction. Without any indication there or comment in the caption, the yields higher 100% are looking a bit odd for the reader on the first view.

Response: The yields above 100% are caused by over-reduction of formamides. Therefore, the maximum yield of amine **3** should be 200%. To clarify this point, we added the following sentence to the captions of Figs. 2 and 3 “Yield of amine **3a** was reported based on the mole of **1a**, being 200% at maximum.”

Comment 5: Line 186-187, lines 200-201 as well as figures 3B, S6 and S15: Generally, I would recommend a in dept computational study on the mechanism, which could help to evaluate, which is the most likely pathway. But this would require a lot of computational work and would be too much for a revision, but I recommend addressing this in a follow-up work. Nevertheless, I strongly recommend for this manuscript to also make the NBO analysis for the methyl-substituted urea and cyclic urea in S6 as well as the ester and amide in figure 3B. If in all the other cases the electron density on the O is lower than in **1a**, this would be a strong support for the metal-ligand cooperativity pathway. On the other hand, as the methylated urea **1o** in scheme S6 is not really working, this is also an indication for the isocyanate path, as this substrate can't form the isocyanate at all (but It could do so, but is also not working).

Response: We agree with the reviewer. Theoretical calculations help reveal the actual reaction pathway by comparing different candidate pathways, in this case metal-ligand cooperative hydrogenation of urea vs. thermal

decomposition of urea. Because such calculation would require a great deal of effort, we plan to report our theoretical calculations on this catalysis in a separate paper in due time.

According to the reviewer's comment, we performed NBO analysis of ester and amide (see Figure IV). Ethyl benzoate (**13**), which was tolerated in the present catalysis, has a less negative NBO charge on its oxygen atom but a comparable electrophilicity on its carbonyl carbon to that of urea **1a**, suggesting the importance of basicity of the carbonyl oxygen. Benzamide **1k'** as a model of urea **1k** that possesses tertiary benzamide moiety and the corresponding secondary benzamide **1k''** have high basicity comparable to urea **1a**. The absence of reaction on the benzamide moiety in the hydrogenation (Fig. 3A, **1k**) can be explained as follows. The basic oxygen atom may become protonated, but the subsequent or concomitant hydride transfer from Ir to the relatively less positive carbonyl carbon is unfavorable in this case compared to urea **1a**. These computational results and discussions were added to the revised Supplementary Information (Section 1-8).

Figure IV. NBO analysis of carbonyl compounds

In contrast, NBO analysis of **1o** and **1t** failed to provide a reasonable explanation why these substrates did not undergo the hydrogenolysis (see Figure V). Compared to **1a**, 1,3-dimethyl-1,3-diphenylurea (**1o**) has a lower NBO charge on the O atom, whereas cyclic urea **1t** has a higher NBO charge at this site. These NBO analysis results may suggest that another factor such as steric hindrance may affect the hydrogenolysis of these substrates. In the case of **1t**, another possibility is the recyclization of the product, *N*-aminoethylformamide.

Figure V. NBO analysis of inapplicable urea substrates

To explain the results above, the sentences in the Supplementary Information were modified as follows:

“In natural bond orbital (NBO) analysis, a large negative value indicates a higher electron density and a large positive value indicates a higher electron affinity. We performed NBO analysis for 1,3-diphenylurea (**1a**), formanilide (**2a**), ethyl benzoate (**13**), and benzamides **1k'** and **1k''** as models of **1k** (Fig. S15). The carbonyl oxygen of **1a** was determined to have the highest electron density (-0.655) compared to **2a**, ester **13**, and benzamides **1k'** and **1k''**. Similarly, the electrophilicity of the carbonyl compounds was estimated by NBO analysis. Although urea has resonance forms with two nitrogen atoms, the carbonyl carbon has a relatively high NBO value (+0.792) compared to other carbonyl functionalities calculated.

Metal-ligand cooperative pathway: Because urea has a higher electron density than formamide, NBO analysis supports the proposed origin of the chemoselectivity based on the basicity of the oxygen atom in the protonation step. The observed functional group tolerance can also be explained by NBO analysis. Compared to urea **1a**, although ester **13** has a comparable electron affinity at the carbonyl carbon (+0.792 (**1a**) vs. +0.786 (**13**)), the basicity of its ester oxygen is much weaker (−0.655 (**1a**) vs. −0.619 (**13**)). Therefore, protonation by pyrrole preferentially occurs at the urea oxygen. Tertiary and secondary benzamides **1k'** and **1k''** have a comparable electron densities at the carbonyl oxygen to that of urea **1a** (−0.655 (**1a**) vs. −0.648 (**1k'**)/−0.653 (**1k''**)). However, the electron affinity of the carbonyl carbon in **1k'** and **1k''** is much weaker than that of **1a** (+0.792 (**1a**) vs. +0.664 (**1k'**)/+0.653 (**1k''**)). Therefore, the hydride transfer to benzamides is unfavorable. These electronic properties of the carbonyl moiety well explain the observed chemoselectivities.”

Comment 6: Line 207-226 and figure 5: The authors used here conditions B from figure 3 which did result in significant overreduction (24% of 19) and therefore selectivity loss. As given in table S5, also reduction without added bases was tried (entry 1 and 2 in table S5), ut did not result in any product. Did the authors have any idea why, as in figure 3D, reduction worked also well with most methylated ureas under conditions A without the base. Also, it would be good to test at least some more polyureas, especially the more common ones with the non-methylated urea units (see for example the polyureas as given in ref 38 but also see: Polyureas, Ullmann’s Encyclopedia of Industrial Chemistry, 2016, DOI: 10.1002/14356007.d21_d01.pub2). In terms of a proper chemocycling, the given example is not the most suitable one du to the 3:1 mixture of bisformamide : formamide obtained. Investigation of the hydrogenation performance on 2-3 other non-methylated polyureas would be very interesting for the reader in terms of polymer recycling.

Response: As pointed out by the reviewer, hydrogenolysis of polyurea **17** required KO^tBu additive, and otherwise no monomeric degradation products were obtained (entries 1-2, Table S5). This result is consistent with the case of **1as**, in that a urea consisting of aniline and aliphatic secondary amine requires the addition of KO^tBu for smooth hydrogenolysis unlike the cases of 1-methyl-1,3-diphenylurea derivatives such as **1ao** (Fig. 3D, see Figure VI).

Figure VI. Representative hydrogenolysis of unsymmetric ureas (From Fig. 3D)

For other polyurea resins, we have investigated polyurea **17'** prepared by the reaction of diisocyanate **15** with hexanediamine (see Figure VII). Unfortunately, **17'** as a substrate displays several problems. 1) It has a low solubility and thus hinders molecular weight determination, 2) Due to the unique chemoselectivity of the present catalysis, there are six potential monomeric products (diamines, monoformamides, and diformamides from two diamine segments in the polymer). 3) The yields of monomeric degradation products could not be determined

quantitatively, because of overlapping NMR signals from hexanediamine and its mono- and di-formamides. GC analysis was not applicable either, since the thermal decomposition of urea moiety in the polymer causes an overestimation of amines.

Indeed, all reported hydrogenolysis of polyurea resins yielded diamine(s) and methanol, and there was no complexity like the ones we encountered here. To solve the problems for selective hydrogenolysis will be our next project.

Figure VII. The structure of other polyurea resin we tested.

Reviewers' Comments:

Reviewer #1:

Remarks to the Author:

The authors have been able to address all the comments satisfactorily which has improved the paper even more. The manuscript will be of great interest to the community and the quality and standard of the supporting information are very high. I recommend the manuscript to be accepted in the Nat Commun.

Reviewer #2:

Remarks to the Author:

In this revised manuscript, the authors responded very well to the reviewed comments and changed the manuscript as well as the supporting information accordingly. So, from my perspective, the authors clarified everything and now it's in the form to be recommended for publication in Nature Communications. Just one minor point as given below and easy to address, which should be changed:

With the newly added ref 39, which was published during this manuscript was in revision, the part line 67-73 should be rewritten (will not change the storyline for the rest of the manuscript), to relate correctly to the published systems. As in the new ref 39 the same cat-system is used than in ref 31, it's the same "system" but just applied to more substrates than in ref 31, and have therefore the same issues according selectivity towards the hydrogenation of other carbonyl compounds. My suggestion for this part (or something like this):

"As formamide intermediates are more reactive than urea, urea derivatives are fully reduced to amines and methanol under these catalytic systems, with the exception using a Ru-triphos catalyst to give a mixture of amines and formamides from different diaryl- and dialkylureas [31,39]. In this catalytic system, however, not only urea but also ester, amide, and other carbonyl functionalities were hydrogenated under similar conditions (*vide infra*) [31]. Indeed, urea-selective hydrogenolysis in the presence of more reactive carbonyl functionalities, such as esters, has never been achieved using these catalysts and is believed to be unfeasible [29–39]"

REVIEWERS' COMMENTS

Reviewer #1 (Remarks to the Author):

The authors have been able to address all the comments satisfactorily which has improved the paper even more. The manuscript will be of great interest to the community and the quality and standard of the supporting information are very high. I recommend the manuscript to be accepted in the Nat Commun.

Reviewer #2 (Remarks to the Author):

In this revised manuscript, the authors responded very well to the reviewed comments and changed the manuscript as well as the supporting information accordingly. So, from my perspective, the authors clarified everything and now it's in the form to be recommended for publication in Nature Communications. Just one minor point as given below and easy to address, which should be changed:

Comment 1: With the newly added ref 39, which was published during this manuscript was in revision, the part line 67-73 should be rewritten (will not change the storyline for the rest of the manuscript), to relate correctly to the published systems. As in the new ref 39 the same cat-system is used than in ref 31, it's the same "system" but just applied to more substrates than in ref 31, and have therefore the same issues according selectivity towards the hydrogenation of other carbonyl compounds. My suggestion for this part (or something like this):

"As formamide intermediates are more reactive than urea, urea derivatives are fully reduced to amines and methanol under these catalytic systems, with the exception using a Ru-triphos catalyst to give a mixture of amines and formamides from different diaryl- and dialkylureas [31,39]. In this catalytic system, however, not only urea but also ester, amide, and other carbonyl functionalities were hydrogenated under similar conditions (*vide infra*) [31]. Indeed, urea-selective hydrogenolysis in the presence of more reactive carbonyl functionalities, such as esters, has never been achieved using these catalysts and is believed to be unfeasible [29–39]"

Response: We agree with this comment and modified the sentence as below.

"As formamide intermediates are more reactive than urea, urea derivatives are fully reduced to amines and methanol under these catalytic systems, with the exception using a Ru-triphos catalyst to give a mixture of formamides and amines from different diaryl- and dialkylureas^{31,39}. In this catalytic system, however, not only urea but also ester, amide, and other carbonyl functionalities were hydrogenated under similar conditions (*vide infra*)³¹."